# Duration and key determinants of infectious virus shedding in hospitalized patients with coronavirus disease-2019 (COVID-19)

Jeroen J. A. van Kampen[1✉], David A. M. C. van de Vijver[1], Pieter L. A. Fraaij[1,2], Bart L. Haagmans [1],
Mart M. Lamers [1], Nisreen Okba [1], Johannes P. C. van den Akker[3], Henrik Endeman[3],
Diederik A. M. P. J. Gommers[3], Jan J. Cornelissen[4], Rogier A. S. Hoek [5,6], Menno M. van der Eerden[5],
Dennis A. Hesselink[6,7], Herold J. Metselaar[6,8], Annelies Verbon[9], Jurriaan E. M. de Steenwinkel[9],
Georgina I. Aron[1], Eric C. M. van Gorp[1], Sander van Boheemen[1], Jolanda C. Voermans[1], Charles A. B. Boucher[1],
Richard Molenkamp[1], Marion P. G. Koopmans [1,10], Corine Geurtsvankessel [1,10] &
Annemiek A. van der Eijk[1,10]

Key questions in COVID-19 are the duration and determinants of infectious virus shedding. Here, we report that infectious virus shedding is detected by virus cultures in 23 of the 129 patients (17.8%) hospitalized with COVID-19. The median duration of shedding infectious virus is 8 days post onset of symptoms (IQR 5–11) and drops below 5% after 15.2 days post onset of symptoms (95% confidence interval (CI) 13.4–17.2). Multivariate analyses identify viral loads above 7 $\log_{10}$ RNA copies/mL (odds ratio [OR] of 14.7 (CI 3.57-58.1; $p < 0.001$) as independently associated with isolation of infectious SARS-CoV-2 from the respiratory tract. A serum neutralizing antibody titre of at least 1:20 (OR of 0.01 (CI 0.003-0.08; $p < 0.001$) is independently associated with non-infectious SARS-CoV-2. We conclude that quantitative viral RNA load assays and serological assays could be used in test-based strategies to discontinue or de-escalate infection prevention and control precautions.

[1] Department of Viroscience, Erasmus MC, Rotterdam, The Netherlands. [2] Department of Pediatrics, Subdivision Infectious Diseases and Immunology, Erasmus MC - Sophia, Rotterdam, The Netherlands. [3] Department of Intensive Care, Erasmus MC, Rotterdam, The Netherlands. [4] Department of Hematology, Erasmus MC, Rotterdam, The Netherlands. [5] Department of Pulmonary Medicine, Erasmus MC, Rotterdam, The Netherlands. [6] Erasmus MC Transplant Institute, Rotterdam, The Netherlands. [7] Department of Internal Medicine, Division of Nephrology and Transplantation, Erasmus MC, Rotterdam, The Netherlands. [8] Department of Gastroenterology and Hepatology, Erasmus MC, Rotterdam, The Netherlands. [9] Department of Medical Microbiology and Infectious Diseases, Erasmus MC, Rotterdam, The Netherlands. [10] These authors contributed equally: Marion P. G. Koopmans, Corine Geurtsvankessel, Annemiek A. van der Eijk. ✉email: j.vankampen@erasmusmc.nl

Coronavirus disease-2019 (COVID-19) is a new clinical entity caused by severe acute respiratory syndrome coronavirus 2 (SARS-CoV-2)[1,2]. In particular, persons with underlying diseases, such as diabetes mellitus, hypertension, cardiovascular disease, and respiratory disease, are at increased risk for severe COVID-19, and case fatality rates increase steeply with age[3].

Understanding the kinetics of infectious virus shedding in relation to potential for transmission is crucial to guide infection prevention and control strategies[4]. Long-term shedding of viral RNA has been reported in COVID-19 patients, even after full recovery, putting serious constraints on timely discharge from the hospital or de-escalation of infection prevention and control practices[5–7]. Detection of viral RNA by reverse transcriptase-polymerase chain reaction (RT-PCR) is the gold standard for COVID-19 diagnosis and this technique is used in test-based strategies to discontinue or de-escalate infection prevention and control precautions[8–10]. However, there is no clear correlation between detection of viral RNA and detection of infectious virus using cell culture[5,11,12]. Detection of infectious virus, also called live virus or replication-competent virus, by demonstration of in vitro infectiousness on cell lines is regarded as a more informative surrogate of viral transmission than detection of viral RNA[8–10]. In a COVID-19 hamster model, the window of transmission correlated well with the detection of infectious virus using cell culture but not with viral RNA[13]. Key questions in COVID-19, like in any other infectious disease, are how long a person sheds infectious virus and what the determinants are of infectious virus shedding[5,11,12,14,15].

Two studies reported that infectious virus could not be detected in respiratory tract samples obtained more than 8 days after onset of symptoms despite continued detection of high levels of viral RNA[5,12]. For one patient, infectious virus shedding up to 18 days after onset of symptoms was reported[11]. Shedding of infectious SARS-CoV-2 has not been studied in larger groups of patients nor in patients with severe or critical COVID-19. Here, we show that patients with critical COVID-19 may shed infectious virus for longer periods of time compared to what has been reported for in patients with mild COVID-19. In addition, we show that infectious virus shedding drops to undetectable levels below a viral RNA load threshold and once serum neutralizing antibodies are present, which suggests that quantitative viral RNA load assays and serological assays could be used in test-based strategies to discontinue or de-escalate infection prevention and control precautions.

## Results

We included 129 hospitalized individuals that had been diagnosed with COVID-19 by RT-PCR and for whom at least one virus culture from a respiratory tract sample was available (Table 1). Of these, 89 patients (69.0%) had been admitted to the intensive care and the remaining 40 patients (31.0%) were admitted to the medium care. Mechanical ventilation was only performed at the intensive care (81 or 91.0% of patients). Supplemental oxygen was given to 8 (9.0%) of the intensive care patients and to 35 (87.5%) medium care patients. Thirty patients were immunosuppressed (23%) of whom 19 (14.7%) were nonseverely immunocompromised and 11 (8.5%) were severely immunocompromised.

We tested 690 respiratory samples from the 129 patients for the presence of infectious virus using cell culture and determined the viral RNA load with RT-qPCR (Fig. 1). Infectious SARS-CoV-2 was isolated from 62 respiratory tract samples (9.0%) of 23 patients (17.8%). The median time of infectious virus shedding was 8 days post onset of symptoms (IQR 5–11, range 0–20) and probit analysis showed a probability of ≤5% for isolating infectious SARS-CoV-2 when the duration of symptoms was 15.2 days (95% CI 13.4–17.2) or more (Fig. 2A). The median viral load was significantly higher in

culture positive samples than in culture negative samples (8.14 versus 5.88 $Log_{10}$ RNA copies/mL, $p < 0.0001$) and the probability of isolating infectious SARS-CoV-2 was less than 5% when the viral load was below 6.63 $Log_{10}$ RNA copies/mL (95% CI 6.24–6.91) (Fig. 2B).

For 27 patients, neutralizing antibody titers from 112 serum samples that were obtained on the same day as a respiratory tract sample were available in our diagnostic database (Table 2). The probability of isolating infectious virus was less than 5% when the neutralizing antibody titer was 1:80 or higher (Fig. 2C). In addition to these neutralizing antibody measurements, we performed RT-PCRs to detect SARS-CoV-2 subgenomic messenger RNA in the 112 corresponding respiratory tract samples. Detection of the subgenomic RNAs outlasted the detection of infectious virus (Supplementary Figs. 1 and 2), and predicted poorly if virus cultures were positive (positive predictive value of 37.5%). In addition, quantitative assessment of subgenomic RNA using cycle threshold (CT) values had no added value over measuring viral genomic RNA loads or serological response to predict infectious virus shedding (Supplementary Fig. 3).

Finally, the key parameters were compared using multivariate generalized estimating equations (Table 3). For this, timepoints for which all three data types (RT-qPCR, virus culture and serum neutralizing antibody titer) were available were included ($n = 112$). A viral load exceeding 7 $Log_{10}$ RNA copies/mL, less than 7 days of symptoms, absence of serum neutralizing antibodies and being immunocompromised were all associated with a positive virus culture in univariate analysis. After submitting all these variables into a multivariate analysis, we found that only a viral load above 7 $Log_{10}$ RNA copies/mL and absence of serum neutralizing antibodies were independently associated with isolation of infectious SARS-CoV-2 from the respiratory tract.

## Discussion

In this study we assessed the duration and key determinants of infectious SARS-CoV-2 shedding in patients with severe and critical COVID-19. Such information is critical to design test-based and symptom-based strategies to discontinue infection prevention and control precautions. Both strategies only allow for discontinuation of infection prevention and control precautions after partial resolution of symptoms. Symptom-based strategies use as additional criterion that a certain time interval should have passed since onset of symptoms, while test-based strategies use negative SARS-CoV-2 RT-PCR results as main additional criterion.

The duration of infectious virus shedding found in this study was longer than has been reported previously[5,11,12]. Wölfel and colleagues showed for patients with mild COVID-19 that infectious virus could not be detected after more than eight days since onset of symptoms[5]. Bullard and colleagues obtained similar results, but disease severity was not reported[12]. Shedding of infectious virus up to 18 days after onset of symptoms has been reported for a single case of mild COVID-19[11]. The patients in this study had severe or critical COVID-19 and detection of infectious virus was common after eight days or more since onset of symptoms. For a single patient, infectious virus was detected up to 20 days after onset of symptoms. Higher viral loads have been reported for severe COVID-19 cases compared to mild cases, which may in part explain the longer duration of shedding found in this study[16–20]. Our findings imply that symptom-based strategies to discontinue infection prevention and control precautions should take diseases severity into account. For example, the CDC currently use a minimum disease duration of 10 days in their symptom-based strategy as the statistically estimated likelihood of recovering replication-competent virus approaches zero after ten days of symptoms[8,21]. Based on our findings, a longer disease duration could be considered for severely-ill patients.

High viral RNA loads were independently associated with shedding of infectious virus, but, upon seroconversion, shedding of

**Table 1 Patient characteristics.**

| Characteristic | All | Intensive care | Ward | p value (ICU vs ward) |
|---|---|---|---|---|
| Number[a] | 129 | 89 (69.0%) | 40 (31.0%) | |
| Male | 86 (66.7%) | 65 (73.0%) | 21 (52.5%) | 0.04 |
| Age (median—IQR) | 65 (57–72) | 66 (57–72) | 63 (57–74) | 0.90 |
| Immunocompromised[b] | | | | |
| Moderate | 19 (14.7%) | 10 (11.2%) | 9 (22.5%) | 0.04 |
| Severe | 11 (8.5%) | 5 (5.6%) | 6 (15.0%) | |
| Clinical parameters | | | | |
| Mechanical ventilation | 81 (62.8%) | 81 (91.0%) | 0 | |
| Supplemental oxygen | 43 (33.3%) | 8 (9.0%) | 35 (87.5%) | |
| Died | 14 (10.9%) | 11 (12.3%) | 3 (7.5%) | |
| Duration of illness[c] | | | | |
| Median (IQR) | 18 (13–21) | 18 (13–22) | 15 (12–18) | 0.009 |
| Tests per patient, total (mean per person) | | | | |
| Culture | 690 (5.3) | 601 (6.8) | 89 (2.2) | |
| PRNT | 112 (0.9) | 82 (0.9) | 30 (0.8) | |
| PCR | 688 (5.3) | 599 (6.7) | 89 (2.2) | |

[a]Disease severity classification according to NIH criteria (https://www.covid19treatmentguidelines.nih.gov/overview/management-of-covid-19/): 81/129 (62.5%) critical disease, 43/129 (33.3%) severe disease, 5/129 (3.9%) moderate disease.
[b]Immunocompromised level was scored as described previously[25]. Patients with severe immunosuppression (n = 11): Lung transplantation, or other solid organ transplantation and treatment for rejection within the last 3 months (n = 3); Underlying disease treated with daily corticosteroid dosages (based on prednisone) >30 mg for >14 days and/or immunomodulating biologicals (n = 4); Allogeneic hematopoietic stem cell transplantation within the last 12 months, or allogeneic hematopoietic stem cell transplantation with with graft-versus-host-disease treated with immunosuppressive drugs, or acute leukemia (n = 4). Patients with nonsevere immunosuppression (n = 19): Untreated auto-immune disease or underlying disease treated with immunosuppressive drugs (excluding treatment with daily corticosteroid dosages (based on prednisone) >30 mg for >14 days and/or treatment with immunomodulating biologicals) (n = 10); At least 1 year after solid organ transplantation (excluding lung transplantation) and no rejection (n = 3); Hematological malignancies (excluding acute leukemia and leukemia treated with induction therapy or chemotherapy resulting in neutropenia for >7 days) (n = 4); Other nonsevere immunodeficiencies (n = 2).
[c]As of April 17th 2020. PRNT = plaque-reduction neutralization titer. Respiratory tract samples for virus culture and PCR were obtained from the lower respiratory tract (sputum) on the intensive care unit (538/690 samples, 78%) and from the upper respiratory tract (swabs) on the intensive care unit as well as on the medium care unit (152/690 samples, 22%). A total of 127 out of the 690 respiratory tract samples that were submitted for virus culture (18.4%) were obtained from immunocompromised patients. For categorical variables a two-sided Chi-square test was used and for continuous variables a two-sided student's t-test was used. No adjustments were made for multiple comparisons.

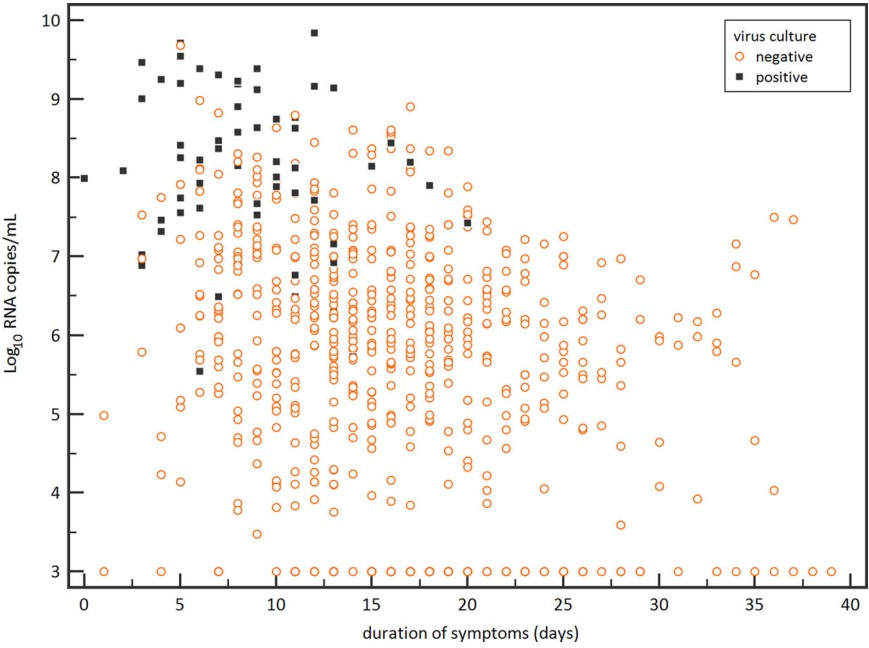

**Fig. 1 Viral loads and duration of symptoms for infectious virus shedding.** Viral RNA loads ($Log_{10}$ RNA copies/mL) in the respiratory samples versus the duration of symptoms (days). Black boxes represent virus culture positive samples and open red circles represent the virus culture negative samples.

infectious virus dropped rapidly to undetectable levels. Infectious virus could not be isolated from respiratory tract samples once patients had a serum neutralizing antibody titer of at least 1:80. These results warrant the use of quantitative viral RNA load assays and serological assays in test-based strategies to discontinue or de-escalate infection prevention and control precautions. The probability of isolating infectious virus was less than 5% when viral RNA load was below 6.63 Log10 RNA copies/mL, which is strikingly similar compared to the cutoff of 6.51 Log10 RNA copies/mL reported by Wölfel et al.[5]. In addition, Bullard and colleagues used cycle threshold (ct) values as quantitative measure for viral RNA load and reported that infectious virus could not be isolated from diagnostic samples when ct values were above 24[12]. Together, these results indicate that viral RNA load cutoffs could be used in test-based strategies to discontinue infection prevention and control precautions. In addition, we report here a very strong association between neutralizing antibody response and shedding of infectious virus with an odds ratio of 0.01 for isolating infectious virus after seroconversion. Antibody responses were measured with a plaque-reduction neutralization test (PRNT)[22]. Neutralization assays, which are the gold standard in coronavirus serology, are labor-intensive and require a biosafety level 3 laboratory. We have recently cross-validated various commercial immunoassays using our PRNT50% as gold standard. Some commercial assays showed good agreement with our PRNT50%: For example, the

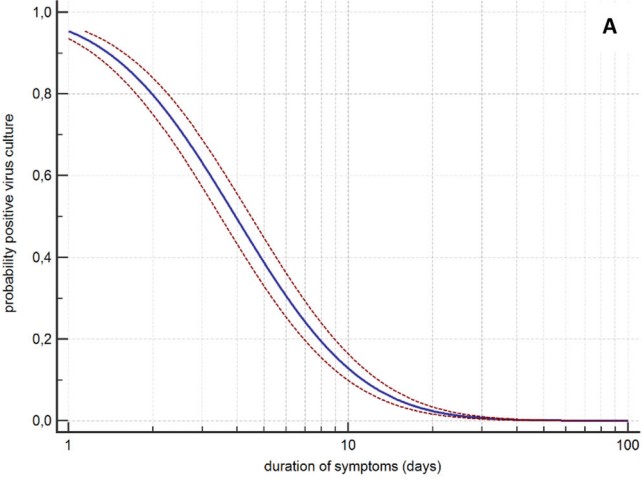

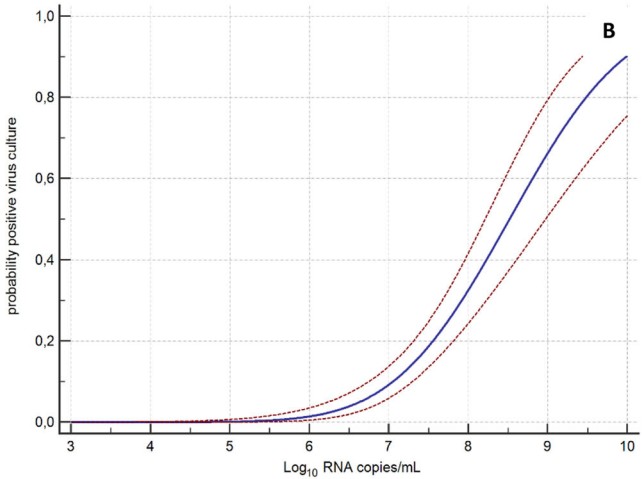

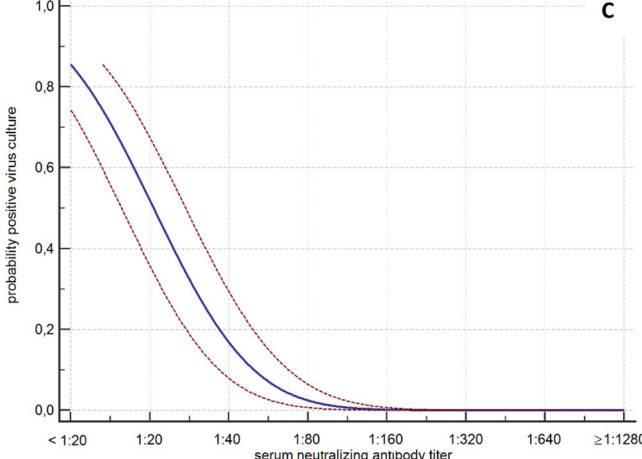

**Fig. 2 Probabilities of infectious virus shedding.** Probit analyses of the detection of infectious virus in respiratory samples with cell culture for duration of symptoms in days (**A**) ($n = 690$ samples), viral RNA load in Log10 copies per mL (**B**) ($n = 688$ samples), and serum neutralizing antibody titer (**C**) ($n = 112$ samples). Blue line represent the probit curve and the dotted red lines represent the 95% confidence interval. Serum neutralizing antibody titers are expressed as plaque-reduction neutralization titers 50% as described previously[27].

**Table 2 Serum neutralizing antibody titers and isolation of infectious virus from the respiratory tract.**

| Serum neutralizing antibody titer | Total number samples | Number culture positive samples (%) | Number culture negative samples (%) |
|---|---|---|---|
| <1:20 | 31 | 27 (87%) | 4 (13%) |
| 1:20 | 10 | 4 (40%) | 6 (60%) |
| 1:40 | 7 | 2 (29%) | 5 (71%) |
| 1:80 | 2 | 0 (0%) | 2 (100%) |
| 1:160 | 4 | 0 (0%) | 4 (100%) |
| 1:320 | 11 | 0 (0%) | 11 (100%) |
| 1:640 | 9 | 0 (0%) | 9 (100%) |
| 1:1280 | 14 | 0 (0%) | 14 (100%) |
| 1:2560 | 16 | 0 (0%) | 16 (100%) |

Serum neutralizing antibody titers against SARS-CoV-2 were determined using a plaque-reduction neutralization assay[17]. Neutralizing antibodies (titers of 1:20 or higher) were detected in 72.3% (81/112) of the serum samples. For six patients, infectious SARS-CoV-2 was isolated from the respiratory tract despite the presence of neutralizing SARS-CoV-2 antibodies in the serum sample pairs. In four of these six patients, infectious virus was not isolated in the consecutive respiratory tract samples obtained after a virus culture positive sample (sampled from day +1, +1, +4, and +4 in respect to virus culture positive sample). For one patient, infectious virus was not isolated in the respiratory tract sample obtained one day after the virus culture positive respiratory tract sample, while the respiratory tract sample obtained 2 days after the virus culture positive respiratory tract sample was positive for infectious SARS-CoV-2. All respiratory tract samples obtained thereafter tested negative for infectious virus. For one patient, no follow-up respiratory tract samples were available.

Wantai SARS-CoV-2 Ig total ELISA has a sensitivity of 99% (95% CI 97–100%) and a specificity of 99% (95% CI 96–100%)[23]. These commercial immunoassays require less stringent biosafety measures and are amenable to high throughput use resulting in a broad application of our results to guide infection prevention strategies and discharge management for clinical cases being hospitalized.

Detection of viral subgenomic RNA correlated poorly with shedding of infectious virus. These RNAs are produced only in actively infected cells and are not packaged into virions. Subgenomic RNAs were still detected when virus cultures turned negative. This could indicate that active replication continues in severely-ill symptomatic COVID-19 patients after seroconversion and after shedding of infectious virus has stopped. Possibly, infectious virions are produced but are directly neutralized by antibodies in the respiratory tract. On the other hand, the half-life of viral subgenomic RNAs is not known in COVID-19 and these RNAs may still be detected once replication has stopped.

Our study has some limitations. Firstly, virological data were obtained from diagnostic samples only and samples were not prospectively collected at predefined timepoints. However, as many aspects of COVID-19 were still unclear, a sampling-rich diagnostic approach was applied in our institution with regular virological monitoring of confirmed COVID-19 patients. This approach resulted in a large high quality dataset from a considerable number of patients including patients with a immunocompromised status. The strikingly similar viral RNA load cutoff for a 5% probability of a positive virus culture found by us and by Wölfel et al. underpins the validity of the results[5]. Secondly, we used in vitro cell cultures as a surrogate marker for infectious virus shedding. The success of SARS-CoV-2 isolation is dependent on which cell lines is used[24]. Vero cells are currently regarded as the gold standard to detect infectious SAR-CoV-2, but the true limit of detection is unknown. Notwithstanding the above, experimental evidence from a COVID-19 hamster model showed that transmission of SARS-CoV-2 correlated well with detection of infectious SARS-CoV-2 from respiratory tract samples using in vitro Vero cell cultures while detection of viral RNA did not[13]. More data from experimental models, and epidemiological and modeling

**Table 3 Univariate and multivariate analysis of key determinants for infectious virus shedding.**

| Variable | Positive virus culture (n = 33) | Negative virus culture (n = 79) | Univariate odds ratio (95% CI) | Multivariate odds ratio (95% CI) |
|---|---|---|---|---|
| Viral RNA load | | | | |
| >10$^7$ RNA copies/mL | 29 (87.9%) | 22 (27.8%) | 18.8 (5.5–64.2), p < 0.001 | 14.7 (3.7–58.1), p < 0.001 |
| Duration of symptoms | | | | |
| <7 days | 20 (60.6%) | 17 (21.5%) | 5.6 (1.7–18.1), p = 0.004 | 2.1 (0.4–11.7), p = 0.31 |
| Serum neutralizing antibody titer | | | | |
| 1:20 or higher | 6 (18.2%) | 75 (94.9%) | 0.01 (0.003–0.05), p < 0.001 | 0.01 (0.002–0.08), p < 0.001 |
| Immunocompromised | | | | |
| Yes | 10 (30.3%) | 10 (12.7%) | 3.00 (0.8–11.0), p = 0.098 | 2.0 (0.7–5.3), p = 0.22 |

Results of the univariate and multivariate generalized estimating equation analysis. The analyses were limited to the samples for which a viral RNA load and a serum neutralizing antibody titer were available from samples taken at the same day.

studies on transmission, which take viral RNA load and antibody response into account, are needed for further validation of this approach. It should be noted that, besides the infectious viral load, additional factors determine virus transmissibility. Finally, our study only included hospitalized symptomatic adults with severe or critical COVID-19 and important differences were noted in our study compared to what has been reported for in mild COVID-19. Thus, further studies are needed on the determinants and duration of infectious virus shedding in specific patient groups.

In conclusion, infection prevention and control guidelines should take into account that patients with severe or critical COVID-19 may shed infectious virus for longer periods of time compared to what has been reported for in patients with mild COVID-19. Infectious virus shedding drops to undetectable levels when viral RNA load is low and serum neutralizing antibodies are present, which warrants the use of quantitative viral RNA load assays and serological assays in test-based strategies to discontinue or de-escalate infection prevention and control precautions.

## Methods

**Samples and patients**. Between March 8, 2020 and April 8, 2020, diagnostic respiratory samples of COVID-19 patients from the Erasmus MC that were send to our laboratory for SARS-CoV-2 PCR were also submitted for virus culture. From these patients, results from SARS-CoV-2 PCRs on diagnostic respiratory samples and results from SARS-CoV-2 neutralizing antibody measurements on serum samples were extracted from our diagnostic laboratory information management system (LabTrain version 3, bodegro, the Netherlands). The following information was extracted from the electronic patient files (HiX version 6.1, ChipSoft, the Netherlands): date of onset of symptoms, disease severity (hospitalized on ICU with mechanical ventilation, hospitalized on ICU with oxygen therapy, hospitalized to ward with oxygen therapy, hospitalized to ward without oxygen therapy), information to classify patients as immunocompetent, nonseverely immunocompromised (excluding diabetes mellitus), or severely immunocompromised as described previously[25], disease severity score according to the NIH classification (https://www.covid19treatmentguidelines.nih.gov/overview/management-of-covid-19/), and whether the patients were still alive or not as of April 17, 2020. Excel 2016 (Microsoft Corp, USA) was used as data collection software.

**Sample processing and analysis**. Swabs from the upper respiratory tract were collected in tubes containing 4 mL virus transport medium (Dulbecco's modified eagle's medium (DMEM, Lonza) supplemented with 40% FBS, 20 mM 4-(2-hydroxyethyl)-1-piperazineethanesulfonic acid (HEPES), NaCO3, 10 μg/ml amphotericin B, 1000 U/mL penicillin, 1000 μg/mL streptomycin). Supernatant was passed through a 45-μm filter and used for PCR analysis and virus culture. For sputum samples, 6 mL sample processing medium (DMEM supplemented with 17 mM HEPES, NaCO3, 1000 U/mL penicillin, 1000 μg/mL streptomycin, 12.5 μg/ml amphotericin B) was added until the final volume was 6 mL. Subsequently, samples were vortexed, centrifuged, passed through a 45-μm filter, and 1 part FBS was added to 1.5 parts supernatant. Subsequently, processed samples were used for PCR analysis and virus culture.

Real-time RT-PCR detection of SARS-CoV-2 was performed using an in-house assay[26] or using the SARS-CoV-2 test on a cobas® 6800 system (Roche Diagnostics). Subsequently, cycle threshold (ct) values were converted to Log$_{10}$ RNA copies/mL using calibration curves based on quantified E-gene in vitro RNA transcripts[5]. SARS-CoV-2 subgenomic RNAs were detected with RT-PCR[5].

Respiratory samples were cultured on Vero cells, clone 118, using 24-wells plates with glass coverslips[27]. Cells were inoculated with 200 μL sample per well and centrifugated for 15 min at 3500 × g. After centrifugation, inoculum was discarded, virus culture medium (Iscove's modified Dulbecco's medium (IMDM; Lonza) supplemented with 2 mM L-glutamine (Lonza), 100 U/mL penicillin (Lonza), 100 μg/mL streptomycin (Lonza), 2.5 μg/mL amphotericin B (department of hospital pharmacy, Erasmus MC), and 1% heat-inactivated fetal bovine serum (Sigma)) was added, and samples were cultured at 37 °C and 5% CO$_2$ for 7 days. Each sample was cultured in triplicate: Two replicates were fixed with ice-cold acetone after 24 and 48 h, respectively irrespective if cytopathic effect (CPE) was visible. The fixed samples were further analyzed with immunofluorescence (see below). The remaining replicate was scored for CPE on a daily basis for 7 days. When CPE was visible, the sample was fixed with ice-cold acetone and further analyzed with immunofluorescence (see below). Virus cultures were regarded as negative if no CPE was visible during 7 days. For immunofluorescence read-out, the fixed cells were washed with phosphate buffer saline (PBS), and incubated for 30 min at 37 °C with 25 μL 1000-fold diluted polyclonal rabbit SARS-CoV anti-nucleoprotein antibodies (Sino Biological, catalogue number 40143-T62). After incubation, samples were washed with three times with PBS and once with deionized water. Subsequently, cells were incubated for 30 min at 37 °C with 25 μL 2000-fold diluted Alexa Fluor 488-labeled polyclonal goat anti-rabbit IgG (Invitrogen, catalogue number A-11070). Subsequently, cells were washed three times with PBS. Finally, cells were incubated for 1 min with 25 μL Evan's Blue (counterstain), washed twice with deionized water, air dried and analyzed with a fluorescence microscope.

Serum neutralizing antibodies titers against SARS-CoV-2 (German isolate; GISAID ID EPI_ISL 406862; European Virus Archive Global #026V-03883) were determined using a plaque-reduction neutralization test[22]. A plaque-reduction neutralization titer 50% (PRNT50%) of 1:20 or more was considered to be positive and a PRNT50% below 1:20 negative.

**Medical ethical approval**. All patient samples and data used in this study were collected in the context of routine clinical patient care. Additional analyses were performed only on surplus of patient material collected in the context of routine clinical patient care. The institutional review board of the Erasmus MC (Rotterdam, The Netherlands) approved the use of these data and samples (METC-2015-306). METC-2015-306 is a generic protocol to study viral diseases. Informed consent for COVID-19 research was waived by the privacy knowledge office of the Erasmus MC (Rotterdam, The Netherlands). Instead, patients had the right to opt-out against the use of their surplus patient material and their medical data for research. The opt-out system of the Erasmus MC was checked for all patients included in this study, and none of the patients included in this study opted-out against the use of their surplus patient material and their medical data for research.

**Statistical analysis**. Categorical and continuous variables were compared using the Chi-square test the student's t-test, respectively. Generalized estimating equations were used to identify factors that are associated with a virus culture positive respiratory tract sample. The continuous data in the generalized estimating equations were dichotomized using various cutoff values. In the main paper we present the results of the best fitting generalized estimating equations using the levels of dichotomizing that had the best fit according to the quasi-likelihood under the independence criterion (QIC)[28]. Sensitivity analysis is shown in Supplementary Table 1 and Supplementary Table 2. All variables having a p value < 0.1 in univariate analysis were submitted into a multivariate general estimating equation to account for repeated measurements obtained from the same patient during hospitalization[29]. For this analysis we used the geepack package version 1.3-1 and R version 4.0.0[29]. Probit analyses were performed with MedCalc version 19.2.3 (MedCalc Software Ltd).

**Role of the funding source**. This work partially was funded through EU COVID-19 grant RECOVER 101003589. The study sponsors were involved neither in the study design, the collection, analysis and interpretation of the data, writing of the report, nor in the decision to submit the paper for publication. The corresponding author had full access to all the data in the study and had final responsibility for the decision to submit for publication.

**Reporting summary**. Further information on research design is available in the Nature Research Reporting Summary linked to this article.

## Data availability

All relevant data are available from the authors upon request. Source data are provided with this paper.

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

## Acknowledgements

We gratefully acknowledge EVA-g and Christian Drosten for provision of the quantified E-gene material. Bibi Slingerland, Ga-Lai Chong, Rose Willemze, Jordy Dekker, George Sips, Stephanie Popping, Daphne Mulders, Alex de Ries, and Jeroen Ijpelaar gratefully acknowledged for their technical and analytical contributions. John T. Brooks (CDC) is acknowledged for his helpful discussions on COVID-19 disease severity classification. This work was partially funded through EU COVID-19 grant RECOVER 101003589.

## Author contributions

J.J.A.v.K. conceived and designed the study, supervised the study, wrote the first draft of the manuscript. D.A.M.C.v.d.V., P.L.A.F., C.G., A.E., M.K., R.M., and C.B. contributed to the conception and design of the study. G.I.A. performed the virus cultures. C.G. and J.J.A.v.K. supervised the virus culture experiments. B.H. and M.M.L. provided intellectual input for virus culture experiments and analyses. N.O. performed the virus neutralization tests. B.L. and C.G. supervised virus neutralization tests. R.M., S.v.B., and J.C.V. supervised and interpreted the molecular analyses. D.A.M.C.v.d.V. performed the statistical analyses. P.L.A.F. and J.J.A.v.K. supervised the clinical data analyses. J.P.C.v.d.A., H.E., D.A.M.P.J.G., J.J.C., R.A.S.H., M.M.v.d.E., D.A.H., H.J.M., A.V., J.E.M.d.S., and E.C.M.v.G. were involved in COVID-19 patient care, data, and specimen collection. All authors discussed the results and implications and commented on the manuscript at all stages.

## Competing interests

The authors declare no competing interests.
