## [Peer Review File · Nature Communications]

Reviewer Comments Initial -

Reviewer #1 (Remarks to the Author):

Jeroen van Kampen and colleagues assessed infectious virus shedding based on samples from hospitalized (severe) cases of COVID-19 pts. In about 18% of pts they could detect infectious SARS-CoV-2 based on virus culture derived from respiratory tract samples. Although the majority of pts did not shed any infectious virus after day 8 (calculated from symptom onset), about 5% of cases were still infectious after 15 days post symptom onset. They identified two predictors of infectious virus: high viral loads above 7 log₁₀ RNA copies per ml and detection of neutralizing antibody titres in the range of at least 1:20. Therefore the main conclusion of this manuscript is that infection prevention strategies should include both quantitative RNA load assays plus serology-based assays.

A PRNT50% was used to define negativity (below 1:20) with respect to infectious viral particles. Were data collected for PRNT90% in parallel? What was the threshold in this assays for infectivity?

Were data assessed by classic ELISA technology (besides use of functional PRNT assays)? Although the latter is preferred from a scientific point of view in order to prove non-infectivity, ELISA assays against spike or nucleocapsid protein of SARS-CoV-2 are more common in clinical routine. In contrast PRNT assays are labor-intensive and require a biosafety level 3 laboratory. Therefore an ELISA surrogate for PRNT assay results would be highly preferable for broad application in order to guide prevention strategies and discharge management for clinical cases being hospitalized.

In an univariate analysis a correlation was described between immunocompromised individuals and positive viral cultures. Did the authors find any correlation of replication competent virus and B-/T-cell based assays (i.e. number of CD3/CD4 T cells, CD8+ memory T cells; immunoglobulin levels or CD19+ B cells counts by flow cytometry etc.)?

Reviewer #2 (Remarks to the Author):

This is an important piece of work which links quantitative SARS-CoV-2 culture with qPCR in the largest cohort to date and importantly includes very ill patients. The analyses are rigorous and the findings of substantial importance for infection control procedure. Several major points warrant addressing:

1) The statistical analysis should include an interaction term between qPCR and serology as effect modification seems very possible given this data. Diagnosis of effect modification (i.e. Ab level impacting infectivity at high viral load only) would be of importance for identifying patients for isolation.

2) The classification of patients as immunocompromised is inadequately described. This category is far too broad and could encompass a huge range of possibilities from mild (diabetes, solid tumor) to extreme (bone marrow transplant). Please provide more detail.

3) The limitation of this being a clinical cohort rather than a specifically designed prospective cohort is acknowledged but not described in sufficient detail. Sample collection is inherently biased towards certain patients in this context and the results are therefore not totally generalizable, even to other critically ill cohorts. As a result, this statement in the discussion is too strong and should be slightly toned down: "These results warrant the use of quant viral load & serologic assays..."

Minor points

1) In the abstract, median duration of shedding should be "median duration of shedding infectious

virus"

2) Define "medium care"

3) Figure 2 legend: what are marker points? Is this actual data?

3) If possible, provide more detail about the cohort: age, co-morbidities, race, gender

2) Parentheses in the abstract are misplaced for the odds ratios

Response to Reviewers

Reviewer 1

Question 1.1: A PRNT50% was used to define negativity (below 1:20) with respect to infectious viral particles. Were data collected for PRNT90% in parallel? What was the threshold in this assay for infectivity? Were data assessed by classic ELISA technology (besides use of functional PRNT assays)? Although the latter is preferred from a scientific point of view in order to prove non-infectivity, ELISA assays against spike or nucleocapsid protein of SARS-CoV-2 are more common in clinical routine. In contrast PRNT assays are labor-intensive and require a biosafety level 3 laboratory. Therefore an ELISA surrogate for PRNT assay results would be highly preferable for broad application in order to guide prevention strategies and discharge management for clinical cases being hospitalized.

Response 1.1: We did not collect PRNT90% data. The validation of our PRNT50% assay has now been published (Okba, *Emerg Infect Dis*, 2020, PMID 32267220). The threshold for positivity of the PRNT50% is 1:20 or higher, which was defined by testing a cross-reactive serum panel from patients recovered from non-CoV respiratory viruses, CMV, EBV, *M. pneumoniae*, seasonal CoVs (229E, NL63, OC43), MERS-CoV, and SARS-CoV (SARS1). Using this threshold for seropositivity, we show in the current manuscript that in patients with a PRNT50% of 1:20 or higher, the odds ratio for detection of infectious SARS-CoV-2 is 0.01. Our PRNT50% assay, using the threshold for seropositivity of 1:20, has been recently been cross-validated against various commercial immuno-assays (Geurtsvankessel et al., *Nature Communications*, 2020, PMID 32632160). Commercial assays included in this cross-validation were: Wantai Ig total ELISA, Wantai IgM ELISA, Euroimmun IgG ELISA, Euroimmun IgA ELISA, DiaSorin XL IgG CLIA, Celles IgM/IgG, InTec IgM/IgG, Orient gene/Healgen IgM/IgG. Some commercial assays showed have a very good agreement with our PRNT50%. For example, using our PRNT50% assay as gold standard, the Wantai Ig total ELISA has a sensitivity of 99% (95% CI 97% - 100%) and a specificity of 99% (95% CI 96% - 100%). To clarify this, we have added the following to the discussion:

“Neutralization assays, which are the gold standard in coronavirus serology, are labor-intensive and require a biosafety level 3 laboratory. We have recently cross-validated various commercial immuno-assays using our PRNT50% as gold standard. Some commercial assays showed good agreement with our PRNT50%: For example, the Wantai SARS-CoV-2 Ig total ELISA has a sensitivity of 99% (95% CI 97% - 100%) and a specificity of 99% (95% CI 96% -

100%).²⁹ These commercial immunoassays require less stringent biosafety measures and are amenable to high throughput use resulting in a broad application of our results to guide infection prevention strategies and discharge management for clinical cases being hospitalized.”

Question 1.2: In an univariate analysis a correlation was described between immunocompromised individuals and positive viral cultures. Did the authors find any correlation of replication competent virus and B-/T-cell based assays (i.e. number of CD3/CD4 T cells, CD8+ memory T cells; immunoglobulin levels or CD19+ B cells counts by flow cytometry etc.)?

Response 1.2: Data from B/T cell assays were not available for our patients. We used data that were collected as part of routine clinical care, and B/T cell assays are not standard of care in our center for patients with COVID-19. It is therefore impossible to assess the correlation between detection of replication competent virus and B/T cell assays.

Reviewer 2

Question 2.1: The statistical analysis should include an interaction term between qPCR and serology as effect modification seems very possible given this data. Diagnosis of effect modification (i.e. Ab level impacting infectivity at high viral load only) would be of importance for identifying patients for isolation.

Response 2.1: We have included the interaction term between qPCR and serology as effect modifier. In summary, we found that the interaction term had a p-value of 0.75 and we therefore did not include the interaction term into the main statistical analysis that was presented in the main paper. We also did not include the interaction term in our multivariate analysis in which we only submitted variables with a $p < 0.1$ in univariate analysis. The interaction term has, nonetheless, been included in Table S1.

Question 2.2: The classification of patients as immunocompromised is inadequately described. This category is far too broad and could encompass a huge range of possibilities from mild (diabetes, solid tumor) to extreme (bone marrow transplant). Please provide more detail.

Response 2.2: In **Table 1** we provide a more details of the immunocompromised classification: “ Immunocompromised level was scored as described previously¹⁶. Patients with severe immunosuppression (n=11): Lung transplantation, or other solid organ transplantation and treatment for rejection within the last 3 months (n=3); Underlying disease treated with daily corticosteroid dosages (based on prednisone) > 30 mg for > 14 days and/or immunomodulating biologicals (n=4); Allogeneic hematopoietic stem cell

transplantation within the last 12 months, or allogeneic hematopoietic stem cell transplantation with graft-versus-host-disease treated with immunosuppressive drugs, or acute leukemia (n=4). Patients with non-severe immunosuppression (n=19): Untreated autoimmune disease or underlying disease treated with immunosuppressive drugs (excluding treatment with daily corticosteroid dosages (based on prednisone) > 30 mg for > 14 days and/or treatment with immunomodulating biologicals) (n=10); At least 1 year after solid organ transplantation (excluding lung transplantation) and no rejection (n=3); Hematological malignancies (excluding acute leukemia and leukemia treated with induction therapy or chemotherapy resulting in neutropenia for > 7 days) (n=4); Other non-severe immunodeficiencies (n=2)."

Please note that the Dutch law does not allow to share specific diagnosis per patient. Therefore, we show aggregated data in a way that the reader is still able to appreciate the level of immune suppression. Diabetes mellitus was not scored as a non-severe form of immunosuppression. We now state that specifically in the methods section.

Question 2.3: The limitation of this being a clinical cohort rather than a specifically designed prospective cohort is acknowledged but not described in sufficient detail. Sample collection is inherently biased towards certain patients in this context and the results are therefore not totally generalizable, even to other critically ill cohorts. As a result, this statement in the discussion is too strong and should be slightly toned down: "These results warrant the use of quant viral load & serologic assays..."

Response 2.3: We have modified the statement in the discussion to "Infectious virus shedding drops to undetectable levels below a viral RNA load threshold and once serum neutralizing antibodies are present, which suggests that quantitative viral RNA load assays and serological assays could be used in test-based strategies to discontinue or de-escalate infection prevention and control precautions. "

Minor remark 1: In the abstract, median duration of shedding should be "median duration of shedding infectious virus"

Response to minor remark 1: we have changed the abstract according to your suggestion.

Minor remark 2: Define "medium care"

Response to minor remark 2: "medium care" is defined as the ward, i.e. a patient admitted to the hospital but not to the intensive care. We provide more information on the COVID-19 disease severity in Table 1 (see our response to minor remark 3).

Minor remark 3: If possible, provide more detail about the cohort: age, co-morbidities, race, gender

Response to minor remark 3: In table 1 we provide information about gender and age. Data about race was not collected for this study. We now provide more information on the immunosuppression score in Table 1 (see our response to question 2 from reviewer 2). In addition, we now provide more information on the COVID-19 disease severity using NIH criteria (www.covid19treatmentguidelines.nih.gov/overview/management-of-covid-19/) in Table 1: “Disease severity classification according to NIH criteria¹⁷: 81/129 (62,5%) critical disease, 43/129 (33,3%) severe disease, 5/129 (3,9%) moderate disease.”

Minor remark 4: Parentheses in the abstract are misplaced for the odds ratios

Response to minor remark 4: we have changed the abstract according to your suggestion.

Reviewer Comments Second Round -

Reviewer #1 (Remarks to the Author):

All of my questions have been answered completely and satisfactorily by the authors. No further comments or suggestions.

Reviewer #2 (Remarks to the Author):

Thank you for addressing both reviewers suggestions.